# The Development of Thematic Core Collections in Cassava Based on Yield, Disease Resistance, and Root Quality Traits

**DOI:** 10.3390/plants12193474

**Published:** 2023-10-04

**Authors:** Caroline Cardoso dos Santos, Luciano Rogerio Braatz de Andrade, Cátia Dias do Carmo, Eder Jorge de Oliveira

**Affiliations:** 1Centro de Ciências Agrárias, Ambientais e Biológicas, Universidade Federal do Recôncavo da Bahia, Cruz das Almas 44380-000, BA, Brazil; 2Embrapa Mandioca e Fruticultura, Nugene, Cruz das Almas 44380-000, BA, Brazil

**Keywords:** *Manihot esculenta* Crantz, germplasm, breeding, genetic resource

## Abstract

Thematic collections (TCs), which are composed of genotypes with superior agronomic traits and reduced size, offer valuable opportunities for parental selection in plant breeding programs. Three TCs were created to focus on crucial attributes: root yield (CC_Yield), pest and disease resistance (CC_Disease), and root quality traits (CC_Root_quality). The genotypes were ranked using the best linear unbiased predictors (BLUP) method, and a truncated selection was implemented for each collection based on specific traits. The TCs exhibited minimal overlap, with each collection comprising 72 genotypes (CC_Disease), 63 genotypes (CC_Root_quality), and 64 genotypes (CC_Yield), representing 4%, 3.5%, and 3.5% of the total individuals in the entire collection, respectively. The Shannon–Weaver Diversity Index values generally varied but remained below 10% when compared to the entire collection. Most TCs exhibited observed heterozygosity, genetic diversity, and the inbreeding coefficient that closely resembled those of the entire collection, effectively retaining 90.76%, 88.10%, and 88.99% of the alleles present in the entire collection (CC_Disease, CC_Root_quality, and CC_Disease, respectively). A PCA of molecular and agro-morphological data revealed well-distributed and dispersed genotypes, while a discriminant analysis of principal components (DAPC) displayed a high discrimination capacity among the accessions within each collection. The strategies employed in this study hold significant potential for advancing crop improvement efforts.

## 1. Introduction

Cassava (*Manihot esculenta* Crantz) plays a vital role in tropical countries as a primary source of carbohydrates. Its significance in the global market has expanded due to its wide range of applications, including starch production for food industries, utilization in non-food industries, an ingredient in animal feed, and bioethanol production [1]. Moreover, cassava serves as a staple food for approximately 800 million people, predominantly residing in Africa, Latin America, and Asia [2].

In Brazil, cassava production reached 18.49 million tons in 2022, cultivated across 1.24 million hectares [3]. With the increasing integration of cassava into the Brazilian agribusiness sector, there is growing concern regarding the availability of improved varieties suitable for diverse growing regions within the country, as well as concern for ensuring an adequate supply of planting material in terms of quantity and timeliness. Substantial progress has been made in research efforts, and in recent decades, there has been a significant increase in the availability of improved cassava varieties in Brazil [4,5].

The genetic progress achieved in cassava breeding heavily relies on effectively harnessing the genetic variability present in germplasm banks (BAG). Accessing the genetic diversity stored in BAGs is crucial for crop improvement, particularly in identifying clones with a high breeding value and superior agronomic performance. Boukar et al. [6] emphasized that the increasing demand for more productive improved varieties, possessing resistance to major pests and diseases, as well as suitability for profitable cultivation and management systems, has placed greater pressure on germplasm banks to enhance their efficiency. Specifically, they need to focus on comprehensive characterizations and germplasm evaluations to identify parent clones with alleles associated with novel traits or alleles capable of improving yield and adding economic value. Nevertheless, conducting extensive characterizations and evaluations across large collections of cassava germplasm is impractical due to the substantial costs and time required to obtain reliable phenotypic data. As a result, utilizing smaller collections that specifically target certain attributes emerges as a highly suitable strategy to optimize and concentrate efforts on the characterizations, evaluations, and utilization of accessions with higher value.

The concept of core collections was Initially introduced by [7] as a strategic approach to represent the maximum variability of an entire germplasm collection through a minimal subset of individuals. This concept serves to address the challenges associated with the management of large collections, which can be burdensome and costly in terms of conservation, cataloging, characterization, and evaluation processes. By carefully selecting and evaluating a small sample of accessions that encompass the genetic and morphological diversity of the entire collection, it becomes possible to enhance the utilization of these genetic resources in breeding programs. However, over time, core collections may expand beyond their original planned size due to the introduction of new accessions. This can perpetuate the issues related to large collections, characterized by high maintenance costs and limited utilization in the development of products and technologies [8]. As a solution, thematic core collections have been proposed to increase efficiency in identifying diversity specifically for targeted and priority traits in breeding programs [9]. By focusing on specific attributes of interest, thematic core collections aim to streamline the identification and utilization of genetic resources, promoting more effective and targeted breeding efforts.

According to Pessoa-Filho et al. [8], large germplasm collections can be organized into several thematic core collections of smaller sizes, each focusing on specific characteristics of interest. This strategy has been increasingly adopted in various genetic improvement programs across different species. Examples include *Olea europaea* L. [10], soy [11], sorghum [12], and chickpeas [13]. Implementing thematic core collections in cassava would yield significant benefits for genetic improvement programs by optimizing the utilization of available resources and reducing redundancy caused by the frequent exchange of germplasm among farmers and the inconsistent naming of the same accession. Recently, Perez-Fons et al. [14] showcased the potential of a basic germplasm collection utilizing genomic and metabolomic information as phenotyping tools. This approach enables the characterization of elite and pre-breeding materials, thereby identifying accessions with high potential to serve as parents in breeding programs, targeting increased tolerance to biotic and abiotic stresses.

Exploring the genetic variations present in germplasm banks is crucial for effectively addressing the challenges and future scenarios related to increased genetic vulnerability to biotic and abiotic stresses [15]. It also plays a significant role in developing product profiles that meet the specific demands of end users and align with the expansion and growth prospects of the cassava species. In this context, extensive efforts have been undertaken to establish core cassava collections, employing diverse methodological approaches. These approaches include the use of molecular markers [16] and, more recently, the integration of quantitative, qualitative, and molecular data, both individually and in combination [17]. These collections encapsulate the broadest phenotypic and molecular diversity among the genotypes within the germplasm bank, while maintaining a limited number of individuals. Nevertheless, thematic collections specifically centered around agronomic attributes have not yet been developed. Therefore, the main objectives of this study were as follows: (i) to establish thematic core collections capable of preserving the maximum possible diversity concerning yield attributes, disease resistance, and root quality, and (ii) to compare the diversity of these thematic collections with that of the entire collection.

## 2. Results

### 2.1. Cassava Thematic Collections

To enhance the efficient utilization of cassava germplasm resources, three distinct thematic collections were meticulously developed. These collections were crafted with a keen focus on the core attributes of root yield (CC_Yield), pest and disease resistance (CC_Disease), and root quality traits (CC_Root_quality). In the CC_Yield collection, we selected clones with lower BLUPs for plant architecture and higher BLUPs for other traits. The CC_Disease collection, on the other hand, focused on genotypes with lower BLUPs for mite severity, as well as for diseases affecting both the aerial and root parts of the plant. In the CC_Root_quality collection, genotypes were chosen for exhibiting lower BLUPs in terms of cyanide content and shorter cooking times.

The initial selection process involved picking 70, 79, and 79 accessions for each respective thematic collection. Then, the selected accessions were evaluated for their redundancy according to identity-by-state (IBS), with the aim of eliminating duplicates among the individuals selected in each thematic collection. Molecular analysis revealed a substantial degree of relatedness (>0.95) among 108 pairs of accessions (Appendix A). This discovery prompted the exclusion of 29 cassava accessions, thereby refining the selection process. In the culmination of this effort, a total of 199 accessions were thoughtfully chosen to constitute the three thematic collections. Within these collections, there were 72 accessions for CC_Disease, 63 for CC_Root_quality, and 64 for CC_Yield. These selections corresponded to 4%, 3.5%, and 3.5% of the individuals encompassing phenotypic information in the entire collection (1810 accessions) (Figure 1).

Regarding the molecular data, 347 accessions were not genotyped. Therefore, the analysis of genetic parameters in the thematic collections was conducted using a subset of 164 genotyped individuals. Among these genotyped individuals, 60 belonged to the CC_Disease collection, 56 belonged to the CC_Yield collection, and 48 belonged to the CC_Root_quality collection.

The selection of accessions for the different thematic collections was characterized by limited coincidence, as these traits were not highly correlated. In most cases, the thematic collections consisted of exclusive accessions. This finding was supported by the Kappa index, which ranged from 0.048 to 0.176 (Table 1). The highest agreement in the accession selection was observed between the CC_Disease and CC_Yield collections, with 11 accessions in common, corresponding to a Kappa index of 0.176. Coincidences between the other collections were lower, with 5 accessions in common between the CC_Root_quality and CC_Disease collections (Kappa index of 0.096), and 2 accessions in common between the CC_Root_quality and CC_Yield collections (Kappa index of 0.11) (Figure 1). The low level of coincidence among accessions in the thematic collections highlights the importance of forming collections with specific and desirable characteristics for the successful improvement of the cassava crop.

### 2.2. Phenotypic Variation in Cassava Thematic Collections

The broad-sense heritability ($h^{2}$) of the quantitative traits used in the establishment of core collections is presented in Appendix A. Most of the traits exhibited moderate to high *h*^2^ values (>0.30), except for the traits frogskin severity, bacterial blight severity, rust severity, marketable and unmarketable fresh root yield, average number of roots per plant, leaf retention, and plant vigor at 1.5 months.

In the development of the CC_Disease thematic collection, individuals displaying minimal symptoms of the evaluated diseases were specifically chosen. As anticipated, there was a noticeable shift in the distribution curve towards reduced mean values and the lower severity of pests and diseases compared to the entire collection, particularly for mite infestation, anthracnose, bacterial blight, brown leaf spot, and blight leaf spot severity (Figure 2). In fact, the average severity score for dust mites in the thematic collection was 2.92, while the overall population mean was 3.03. Similarly, the average scores for shoot diseases in the thematic collections were slightly lower compared to the entire collection: anthracnose had an average score of 1.41 vs. 1.48 in the entire collection, bacterial blight—1.75 vs. 1.80, brown leaf spot—1.56 vs. 1.62, and blight leaf spot—0.42 vs. 0.47.

The attributes associated with the thematic collection CC_Root_quality were primarily chosen with an emphasis on preserving genotypes that exhibited superior culinary qualities (Figure 3). Consequently, there was an observable shift in the curve towards diminishing the hydrocyanic acid content in the roots. This selection primarily focused on clones with scores below 5 in the picrate test, which essentially denoted safe options for human consumption. The average score for hydrocyanic acid content (Ro.HCN) across the entire dataset was 5.79, while the thematic collection displayed an average of 4.08. Concerning the total carotenoid content in the roots (RO.TCC), there was minimal variation in the mean value of the CC_Root_quality thematic collection (2.22) when compared to the complete collection (2.40). Notably, accessions featuring the highest carotenoid values were not chosen for inclusion in this collection. This could be attributed to their potentially elevated cyanogenic compound content, rendering them unsuitable for direct consumption due to associated safety concerns.

Regarding the additional qualitative traits present in the CC_Root_quality thematic collection, there was a notable rise in the occurrence of clones with a cooking time under 30 min. This particular duration is in line with the preferred standard in Brazil for selecting top-fresh consumption clones (sweet clones). As a result, this increase contributed to a heightened prevalence of clones with pulp textures characterized as friable and very friable. These textures are directly linked to an improved cooking capacity. While the average total carotenoid content displayed no apparent variation, the CC_Root_quality thematic collection experienced an upsurge in the frequency of clones featuring cream and yellow-colored pulp.

The thematic collection dedicated to yield attributes (CC_Yield) comprised accessions that held significant importance for the industrial cassava market. Just like the characteristics observed in the other thematic collections, there was a distinct shift in the distribution curves of traits, favoring attributes related to yield. This shift was particularly pronounced for attributes that displayed substantial variation across the entire collection. For instance, the following attributes varied: plant height (216.06 cm in the thematic collection versus 191.67 cm in the entire collection); shoot yield (25.30 t.ha^−1^ vs. 19.55 t.ha^−1^, respectively); marketable root yield (18.08 vs. 12.8 t.ha^−1^, respectively); unmarketable root yield (6.42 vs. 5.48 t.ha^−1^, respectively); total fresh root yield (24.25 vs. 17.52 t.ha^−1^, respectively); dry root yield (7.67 vs. 5.46 t.ha^−1^, respectively); leaf retention (2.34 vs. 1.98, respectively); and dry matter content (36.02% vs. 34.85%, respectively). Furthermore, the vigor of plants within the CC_Yield collection, assessed at both 1.5 and 12 months after planting, displayed higher values in the thematic collection compared to the entire collection. By contrast, characteristics with less variation among cassava accessions in the thematic collection, relative to the entire collection, were highlighted through box plots for cortex thickness, harvest index, and average root length (Figure 4).

For qualitative attributes, the accessions chosen within the CC_Yield collection displayed only minor variations in the distribution of classes. Across most instances, the chosen accessions exhibited traits like easily peelable root skin, roots with minimal constrictions and lacking a peduncle (sessile), as well as either compact or open growth patterns (Figure 4). Overall, these findings suggest that the cassava accessions picked via thematic collections possess attributes that hold significant agronomic value for the species’ enhancement. Beyond their potential for immediate commercial applications, this selection is poised to facilitate the preservation of promising accessions tailored to each collection’s objectives. Moreover, it should enable a more comprehensive assessment of both phenotypic and genotypic traits.

### 2.3. An Analysis of the Phenotypic Diversity of the Thematic Collections

The evaluation of genetic diversity in the developed thematic collections was performed using the Shannon–Weaver index (ISW) (Table 2). In general, the ISW values in the thematic collections showed no significant difference of more than 10% compared to the entire collection. However, there was a notable exception in the thematic collection CC_Root_quality, where the ISW for Ro.CT (cortex thickness) more than doubled. Conversely, the ISW for Ro.CT was null in the CC_Yield and CC_Disease collections, indicating the absence of variation for this trait in the cassava clones selected by these thematic collections.

### 2.4. Genetic Diversity of the Cassava Thematic Collections Using Molecular Markers

Most of the thematic collections exhibited genetic parameters (*Ho*, *Hs*, and *Fis*) that were very similar to those of the entire collection, both in terms of means and range (Table 3). Additionally, the thematic collections successfully retained a significant proportion of the alleles present in the entire collection, with retention rates of 90.76%, 88.10%, and 88.99% for CC_Disease, CC_Root_quality, and CC_Disease, respectively. Regarding the distribution density of the genetic parameters, the most noticeable differences compared to the entire collection were observed in the CC_Root_quality thematic collection, particularly for the *Hs* parameter, which showed a reduction in the class interval between 0.0 and 0.2 (Figure 5). However, for the other parameters, no substantial changes were observed in the distribution pattern of molecular parameters.

### 2.5. The Validation of Thematic Collections

The representativeness and diversity of each thematic collection were evaluated using principal component analysis (PCA), applied to both phenotypic and molecular data. The genetic relationships between the evaluated accessions were examined to infer the phenotypic and genetic structure (Figure 6). The first two principal components of the PCA explained 13.88% and 31.03% of the variation in the molecular and phenotypic data, respectively. The distribution of cassava clones in the PCA plot based on molecular data was broader compared to the phenotypic data, likely due to the larger amount of available data. However, irrespective of the data type, there was no distinct clustering of clones belonging to each specific thematic collection. This suggests that while there are groups of cassava clones with relatively similar phenotypes, sufficient genetic variability is maintained within the thematic collections for productivity, resistance to pests and diseases, and root quality. The distribution of clones in the PCA plot supports the notion that both molecular and phenotypic diversity are preserved in these collections when compared to the entire collection.

Additionally, a discriminant analysis of principal components (DAPC) was conducted to examine the distribution pattern of clones within the thematic collections compared to the entire collection (Figure 7). By selecting 500 principal components (PCs) and three discriminant functions (DFs), we were able to explain 88% of the total observed genetic variance. With the thematic collections defined as a priori groupings, DAPC successfully discriminated cassava accessions based on their thematic collection membership. The results showed a clear separation of the three thematic collections in the DAPC plot, with only a few instances of overlap between the CC_Disease and CC_Yield collections. Notably, the CC_Root_quality collection appeared to be the most distinct among the three thematic collections.

## 3. Discussion

### 3.1. The Formation and Diversity of the Cassava Thematic Collections

Cassava breeding programs rely on effectively harnessing the genetic variability preserved in germplasm banks to identify suitable parents for producing high-performing clones. However, the selection, conservation, and comprehensive characterization of parents pose significant challenges for many cassava breeding programs, given the vast number of accessions stored in germplasm banks and the extensive array of hybrids generated by breeding programs.

To address this issue, the concept of core collections, which are representative subsets of accessions capturing the genetic diversity of a species, has been employed. However, the increasing size and number of collections in germplasm banks have necessitated the implementation of strategies to constrain collection sizes. Core collections, in some cases, may still include a large number of accessions, thereby perpetuating the challenge they were designed to alleviate. Ideally, core collections should consist of a manageable number of accessions with sufficient genetic variability to be effectively utilized in genetic breeding programs [8].

Conversely, thematic collections, comprising distinctive accessions possessing specific manageable traits, have emerged as a proposed strategy within breeding programs to tackle challenges related to collection size and alignment with breeder preferences [8]. In alignment with this approach, this study identifies cassava thematic collections designed to safeguard and preserve accessions endowed with valuable traits pivotal for creating cultivars that possess desired product profiles. In this pursuit, we have established thematic collections that focus intently on traits of elevated agronomic significance for the cassava crop. Our objective revolves around conducting comprehensive agronomic characterizations, targeting attributes crucial for commercial viability in both table and industrial cassava applications. Furthermore, these collections are strategically positioned as repositories for genetic resources earmarked for strategic breeding endeavors. Through planned crosses, they serve as valuable reservoirs, enabling the derivation of progeny and the determination of breeding values for these select clones.

To fulfill the objective of preserving accessions with shared specific characteristics, while also possessing divergent phenotypic traits and alleles, it is crucial to eliminate potential redundancies within thematic collections [18]. Understanding the genetic relationships among selected accessions is a crucial initial step in assessing the potential utility of germplasm resources [19]. In line with this, an IBS test was conducted to identify 29 accessions exhibiting a high degree of genomic relatedness. These accessions were promptly excluded from their respective thematic collections to ensure the preservation of genetic diversity and to avoid redundancy.

Duplicate accessions within cassava germplasm banks are a common issue that can hinder progress in the species’ improvement [18,20,21]. Recognizing this, redundancy analyses were conducted to address this concern. As a result, the final selection of accessions for the thematic collections focused on yield attributes (CC_Yield), resistance to pests and diseases (CC_Disease), and root quality (CC_Root_quality) yielded 64, 72, and 63 accessions, respectively. These numbers are considered adequate for the capture of full diversity, encompassing both phenotypic and allelic variations, while still facilitating subsequent evaluations and field validations [16]. By eliminating duplicate accessions and ensuring the integrity of the collections, research can now proceed with improved efficiency and accuracy in the pursuit of cassava improvement.

In the context of soybean (*Glycine max*), Gou et al. [11] successfully developed thematic collections with a similar number of genotypes to our study, focusing on traits such as tolerance to cold, drought, salinity, resistance to soybean cyst nematode and soybean mosaic virus, as well as high protein and fat content. Their approach involved comprehensive agronomic and nutritional assessments. A comparison was made between the soybean thematic collection and a secondary mini-collection formed using microsatellite markers (SSR). The findings revealed that the soybean thematic collection exhibited comparable levels of molecular diversity to the mini-collection. However, at the phenotypic level, the thematic collection consisted of soybean accessions with more desirable traits compared to the mini-collection [11]. These results highlight the effectiveness and potential advantages of employing thematic collections in capturing and preserving valuable genetic resources with specific traits for targeted breeding programs.

While thematic collections are primarily focused on gathering accessions with specific characteristics rather than maximizing phenotypic or genotypic diversity, we conducted a comparative analysis of various genetic parameters with the entire collection. Through the analysis of molecular markers, the thematic collections CC_Disease, CC_Root_quality, and CC_Disease captured 90.76%, 88.10%, and 88.99% of the alleles present in the entire collection, respectively, indicating a substantial representation of alleles. Similar outcomes have been observed in other species, such as wild rice (*Oryza rufipogon* Griff.), where 90% of the alleles from the entire collection were retained in a selection of 130 accessions to form a central collection for wild germplasm, facilitating an ex situ conservation of the species [22]. Preserving maximum allelic variation in thematic collections holds strategic significance for maintaining long-term variability, especially since many traits of agronomic importance, including disease resistance, often follow a simple Mendelian inheritance pattern [23]. This underscores the importance of retaining diverse alleles within thematic collections to ensure the availability of valuable genetic resources for future breeding and conservation efforts.

Another parameter utilized for evaluating the excellence of thematic collections is the ISW, a measure encompassing allelic richness and evenness within a given sample [24]. Our study revealed minimal discrepancies between the thematic collections and the complete collection, with variations amounting to less than 10%. This outcome underscores the robust retention of allelic diversity within the thematic collections, maintaining a harmonious distribution of phenotypic classes. Notably, the ISW has proven to be a valuable criterion in other studies for the purpose of selecting subsets that exhibit optimal allelic coverage and genetic distance [10].

A germplasm collection aiming to enhance root quality must prioritize accessions displaying favorable culinary and nutritional attributes. A significant limiting factor for the fresh consumption of cassava is the presence of elevated levels of cyanogenic compounds. Hence, our thematic collection CC_Root_quality screened individuals based on the picrate method classification scale [25] to include only those with a score <5 for cyanogenic compound content, commonly referred to as sweet cassava suitable for cooked consumption [26]. Another important consideration in accession selection for this collection was the carotenoid content of the roots. Beyond imparting a desirable yellow color on the roots, particularly valued in many regions of Brazil, these compounds function as potent antioxidants. Their conversion into provitamin A holds the potential to address hidden hunger arising from vitamin deficiencies, thereby contributing to the cassava biofortification initiative [27]. Notably, despite the concerted effort to prioritize accessions with elevated carotenoid levels and commendable culinary traits, this pursuit encountered challenges in the CC_Root_quality collection. The intersection of high carotenoid content with elevated cyanogenic compounds prevailed among most accessions. In fact, de Carvalho et al. [28] identified a modest yet positive correlation (0.23) between the cyanogenic compound and carotenoid content in cassava roots. Nevertheless, the CC_Root_quality collection effectively amplified the representation of clones characterized by a cream or yellow root pulp. Such attributes hold significance in markets that preferentially seek roots with a non-white pulp, regardless of carotenoid content. Hence, we assert that the selections made within the CC_Root_quality collection impeccably align with the objectives of the breeding program.

A notable feature of the CC_Root_quality thematic collection is the reduced cooking time of the cassava roots. A significant portion of the evaluated accessions demonstrated cooking times of under 40 min. Our collection has intentionally focused on selecting accessions with cooking times of less than 25 min, along with accessions showcasing easily crumbly or finely crumbly pulp after cooking. These two attributes stand as paramount qualities for achieving widespread commercial acceptance in both Brazil and various African countries. This preference is rooted in the fact that consumers primarily seek sweet cultivars that not only cook swiftly but also break down effortlessly during the cooking process [29,30].

The CC_Yield collection exclusively focuses on the economic and industrial aspects that are crucial for the success of the cassava crop in the market, particularly in processing agroindustries involved in flour or starch production [1,31]. In this thematic collection, the selection of accessions was primarily based on key traits that contribute to higher productivity in terms of shoot and fresh and dry root yield. Moreover, characteristics that enhance efficiency in industrial processing were also prioritized, such as easily peeled roots, smooth roots with minimal or no constrictions, and the presence of a peduncle. Additionally, compact plant architecture was considered to enable higher planting densities, resulting in an increased number of plants per unit area. These selection criteria ensure that the CC_Yield collection encompasses accessions with desirable attributes essential for meeting the demands of processing industries and maximizing economic returns from cassava cultivation.

The genotypes chosen for the CC_Yield collection exhibited promising traits that make them ideal candidates for population improvement, considering the presence of alleles associated with increased fresh root yield, higher dry matter content, and elevated starch content. These attributes are of paramount importance in cassava selection programs. However, it is crucial to note that these traits are influenced by non-additive genetic effects, as highlighted by [32]. Therefore, it becomes imperative to employ selection methods that account for these effects in population improvement and the identification of genotypes for release as cultivars. In this context, the advancement of selective processes such as genomic selection can play a pivotal role in enabling the early selection of superior parents and clones for further breeding advancements [32]. By incorporating genomic selection strategies, breeders can enhance the efficiency of selection and expedite the development of high-yielding cassava cultivars with improved agronomic attributes.

The CC_Disease thematic collection was specifically designed to enhance resistance against pests and diseases. To achieve this goal, accessions were selected based on their minimal severity ratings for various issues, including mite severity, shoot diseases like anthracnose, bacterial blight, brown leaf spots, blight leaf spots, rust, and root rot. These pests and diseases have a detrimental impact on cassava crops, leading to decreased photosynthetic rates, stem damage, reduced propagation material for subsequent planting, and diminished root and starch productivity [33]. In fact, losses exceeding 75% in root yield have been documented due to bacterial blight and anthracnose [34]. Root rot is particularly destructive and challenging to control, often resulting in complete crop losses and significant economic damage, since it directly affects the commercial product, which is the root of cassava. Although there is limited documentation on economic losses caused by leaf spots and rust, field observations indicate a reduction in starch productivity by approximately 20–30% due to severe leaf drop and subsequent starch reallocation from roots to rejuvenate the aerial parts of the plant. Thus, the CC_Disease collection encompasses genotypes with the potential for resistance against the major pests and pathogens affecting cassava, providing an efficient and farmer-preferred approach to disease control.

There is limited literature addressing thematic collections associated with cassava diseases. One of the rare instances in cassava research was carried out by Perez-Fons et al. [14], who dedicated their efforts to establishing a core collection with a focus on enhancing biotic stress tolerance in cassava. In this context, specific traits and metabolites were identified, some of which have the potential to confer intrinsic whitefly tolerance. Conversely, in other species, there are numerous examples of thematic collection formation. Silva et al. [35] developed a thematic collection comprising accessions intended for the preemptive breeding of *Fusarium* wilt resistance in banana. Multiple sources of resistance to *Fusarium* wilt and sterility mosaic disease were identified in the core collection of cowpea, both in field and greenhouse conditions [12]. In all of these studies, the establishment of collections facilitated the selection of resistant accessions that will play a significant role in the genetic improvement programs of their respective crops.

### 3.2. A Comprehensive Representation of Genetic Variation

PCA conducted on the complete cassava collection revealed a complex population structure, particularly in the analysis of molecular data. However, when examining both the phenotypic and molecular data through PCA, no distinct clusters or groupings were observed based on the thematic collections to which the accessions belonged. This suggests that while the accessions within each thematic collection share common and desirable characteristics, they exhibit significant phenotypic and genotypic discrepancies. These findings highlight the need for further investigation into the underlying factors contributing to the observed variations within thematic collections and the potential implications for cassava breeding programs.

In contrast to PCA, use of the DAPC in the classification of cassava accessions in each thematic collection, as a priori information, resulted in a more distinct separation of accessions based on their thematic collections. DAPC, as described by [36], aims to maximize diversity between groups while minimizing diversity within groups through discriminant functions. By plotting individuals and groups in a two-variable space, DAPC effectively highlights differences between groups, unlike PCA, which primarily focuses on total variance along directions. The limitations of PCA in discriminating between groups becomes evident, while DAPC successfully reveals the differences between thematic collections [36]. Consequently, the CC_Root_quality collection exhibited almost complete separation, while a slight overlap was observed between the CC_Disease and CC_Yield collections. Notably, DAPC demonstrates robustness across different organisms, independent of ploidy levels and genetic recombination rates, making it a versatile technique for detecting substructures. The sensitivity of DAPC in identifying substructure within cassava hierarchical models has previously been demonstrated by [21], who successfully distinguished distinct groups of duplicate accessions. The application of DAPC in our study reveals a more nuanced differentiation between cassava accessions belonging to different thematic collections, shedding light on the underlying population structure and aiding in the characterization of genetic diversity within and between collections.

### 3.3. Unlocking Practical Utility: The Significance of Thematic Cassava Collections

The objectives of cassava breeding programs are determined based on the specific requirements of different regions of recommendation, which often involve diverse cultural attributes. However, certain common goals, such as enhancing productivity, improving root quality, and enhancing resistance to pests and diseases, transcend regional boundaries. With this in mind, this study aimed to construct thematic collections centered around these essential themes, acknowledging their universal relevance in cassava breeding programs.

With the aim of providing a foundational resource for selecting parental lines that can be utilized in population improvement, associative mapping, genomic selection programs, and advanced studies of the transcriptome, methylome, and epigenomics, our results hold the potential to deliver rapid and targeted impacts for breeders and germplasm curators. These thematic collections serve as an excellent foundation for initiating population improvement efforts aimed at increasing the frequency of favorable alleles for specific agronomic traits. Recurrent selection, for instance, is recommended for developing superior populations, particularly for traits controlled by polygenic inheritance, which is the case for many of the characteristics utilized in the construction of cassava thematic collections.

In addition to population improvement, these thematic collections provide valuable resources for conducting associative mapping studies in germplasm or biparental populations, aiming to identify quantitative trait loci (QTLs) of interest. Associative mapping offers an alternative to family-based linkage mapping by associating allelic variations of agronomically important traits with molecular markers in the collections of unrelated individuals [37,38]. The identification of causal genes associated with specific phenotypic traits is a fundamental objective in plant breeding programs. In the context of cassava, associative mapping has been successfully employed to identify QTLs for carotenoid content, serving as a basis for the biofortification program [39]. Using associative mapping, Wolfe et al. [40] elucidated the genetic architecture that underlie resistance to the African cassava mosaic virus, caused by multiple geminiviruses of the genus *Begomovirus* (Family Geminiviridae). In the context of pest resistance, [41] identified resistance alleles against the green spider mite (*Mononychellus tanajoa* (Bondar)) in African germplasm, utilizing a diversity panel comprising 845 improved cassava genotypes.

The establishment of thematic collections and the screening of accessions with desirable traits related to root quality, yield, and disease resistance in this study will greatly facilitate associative mapping studies and potentially pave the way for future genomic selection investigations. By simultaneously predicting the genetic effects of markers, it becomes possible to capture the effects of a large number of loci and explain a significant portion of the genetic variation underlying quantitative traits. This research represents an initial step in a broader initiative aimed at characterizing the genetic foundation and maximizing the utilization of available cassava germplasm. Consequently, our work holds significant potential to contribute to national and international breeding programs, particularly those focusing on enhancing productivity and disease resistance traits.

Comprising carefully selected accessions tailored to each specific theme and deliberately designed to be compact in size, the thematic collections developed in this study successfully fulfill the initial objective of facilitating the management and utilization of the most valuable and accessible accessions within each germplasm set. Furthermore, concentrating efforts on these thematic collections will streamline the comprehensive characterization of all relevant descriptors for the cassava species, thereby minimizing redundancies among accessions and enhancing the likelihood of identifying rare and valuable alleles for species improvement. This focused approach maximizes efficiency and optimizes the chances of making significant advancements in cassava breeding and germplasm enhancement.

## 4. Materials and Methods

### 4.1. A Comprehensive Data Collection of the Morpho-Agronomic Traits

The study encompassed a collection of 1665 cassava accessions sourced from the germplasm bank and the breeding program of Embrapa Mandioca e Fruticultura, located in Cruz das Almas, Bahia, Brazil (Latitude: 12°40′19″ S, Longitude: 39°06′22″ W, and Altitude: 220 m). The accessions were from various regions of Brazil, and some were obtained through exchange with countries such as Colombia, Venezuela, Nigeria, Mexico, and Uganda. The collection contained local and improved varieties obtained through breeding techniques, including crossings, mass selection, and identification by producers or research institutions. To establish thematic core collections, specific descriptors were evaluated based on their relevance to yield attributes, disease resistance, and root quality, as outlined by Fukuda et al. [42]:

#### 4.1.1. Yield Traits

Plant height (Pl.H), measured from the ground to the plant’s meristem;Plant architecture (Pl.A), evaluated using a 1–5 rating scale: 1 = excellent architecture (no branching or erect stems), 2 = good architecture (branching above 1.60 m or low branching with at least 1.6 m of erect stems), 3 means moderate architecture (branching above 1.20 m or low branching with at least 1.2 m of erect stems), 4 = poor architecture (branching above 0.80 m or low branching with at least 0.80 m of erect stems), and 5 = very poor architecture (highly branched clones with less than 0.80 m of erect stems);Plant type (Pl.T), assessed using a 1–5 rating scale: 1 = compact, 2 = open plant, 3 = umbrella-like, and 4 stands for cylindrical;Shoot yield (ShY), measured in t.ha^−1^ as the total weight of above-ground parts, including leaves, petioles, and stems;Marketable fresh root yield (C.FRY), measured in t.ha^−1^ and characterized by roots with an absence of pest and disease symptoms and the standard size and shape of the genotype;Unmarketable fresh root yield (NC.FRY), measured in t.ha^−1^ as the rest of C.FRY;Total fresh root yield (T.FRY), assessed as C.FRY + NC.FRY in t.ha^−1^;Dry matter content (DMC), expressed as a percentage and measured using the hydrostatic balance method [43];Starch content (StC), expressed as a percentage and measured by extracting starch from the roots [44];Dry root yield (DRY), measured in t.ha^−1^ and obtained by multiplying T.FRY and DMC;Harvest index (HI), assessed as T.FRY/(T.FRY + ShY);Cortex thickness (Ro.CT), measured in millimeters (mm);The presence of a peduncle (Ro.Pe), assessed on a scale where 0 = sessile, 3 = pedunculated, and 5 = a mixture of sessile and pedunculated roots;Root constriction (Ro.Co), assessed using a 1–3 rating scale: 1 = few or no constrictions, 2 = some constrictions, and 3 = many constrictions;Average root length (Ro.Le), measured in centimeters (cm) from the base to the tip;Average root diameter (Ro.Di), measured in millimeters (mm) at the central part of the roots;Easy root peeling (Ro.EP), evaluated at harvest time on a rating scale: 3 = easy peeling, 5 = medium peeling, and 7 = difficult peeling;Leaf retention (Pl.LR), assessed using a rating scale: 1 = less than 5% leaf retention, 2 = 6–15% leaf retention, 3 = 16–30% leaf retention, 4 = 31–50% leaf retention, and 5 = >50% of leaf retention;Plant vigor at 1.5 months (Pl.V1.5M), assessed using a rating scale: 1 = low vigor, 3 = intermediate vigor, and 5 = high vigor;Plant vigor at 12 months (Pl.V12M), similar to Pl.V1.5M;The average number of roots per plant (Ro.NP).

#### 4.1.2. Resistance to Pests and Diseases

Mite severity (P.MS), rated as follows: 0 = no damage; 1 = some white–yellow spots extending to the base of leaves on the apical shoot; 2 = moderate yellow spots on all leaves; 3 = abundant spots on middle-third leaves, slight apical shoot deformation; 4 = severe leaf deformation on apical shoot, whitish appearance, some defoliation, and stem with yellow spots; 5 = highly twisted apical shoots /severe stem spots; 6 = dry apical shoots, leafless, or dead;Anthracnose severity (*Colletotrichum* sp.) (D.AntS), rated as follows: 1 = no symptoms; 2 = small or old cankers on the lower half of the plant; 3 = deep cankers on the upper half of the plant; 4 = deep cankers with sporulation, leaf distortion, wilting, and apex drying; 5 = apical death or plant death;Bacterial blight severity (*Xanthomonas axonopodis* pv. *manihotis*) (D.CBB), scored as follows: 1 = no symptoms, 2 = only foliar symptoms (angular leaf spot), 3 = dark necrotic lesions on stems or petioles, 4 = severe leaf symptoms and/or necrotic lesions with gum exudation, 5 = complete leaf loss with apical death or plant death;Brown spot severity (*Passalora henningsii*) (D.BrLS), scored as follows: 0 = no symptoms; 1 = mild symptoms on the lower third leaves; 2 = leaf spots on lower leaves and yellowing on a few affected leaves; 3 = leaf spots on lower and middle leaves, yellowing on most affected leaves; 4 = leaf spots across the entire plant, lower leaf shedding; 5 = complete plant defoliation;Blight leaf spot severity (*Passalora viçosae*) (D.BlLS), evaluated similar to D.BrLS;Rust severity (*Uromyces manihotis*) (D.RS), scored as follows: 0 = no pustules or damage on stems and leaves, 1 = a few pustules without branch drying, 2 = many pustules with branch drying;Frogskin severity (virus and phytoplasma complex) (D.FS), assessed as 0 = no symptoms on roots; 1 = few roots with corky epidermis 2 = many roots with a corky epidermis, deep cracks, or absence of tuberous root production;Root rot severity (*Phytophthora* sp., *Pythium* sp., *Phytopythium* sp.), dry Rot (*Fusarium* sp.), and black rot (*Neoscytalidium hyalinum* and *Lasiodiplodia* sp.) (D.RRS), rated as follows: 0 = no root rot, 1 = roots up to 25% with root rot, 2 = roots with more than 26% having root rot.

#### 4.1.3. Root Quality Traits

Root cooking time (Ro.CoT), measured at intervals of 15, 20, 25, 30, and 40 min, according to [45];Root Pulp Color (Ro.PC), categorized as follows: 1 = white, 2 = cream, 3 = yellow, 4 = pink, 5 = orange;Root skin color (Ro.SC), rated as follows: 1 = white, 2 = yellow, 3 = light brown, 4 = dark brown;Root cortex color (Ro.CC), scored as 1 = white or cream, 2 = yellow, 3 = pink, and 4 = purple;Cooked root friability (Ro.RF), which is friability evaluated after cooking, based on [46]: 1 = very cohesive, no cracks; 2 = cohesive roots with a cracked center; 3 = friable, with the center cracked and surface partially cracked; 4 = highly friable, with extensive cracking in both center and surface;Hydrogen cyanide content (Ro.HCN), assessed using the picrate method as detailed by [25];Total carotenoid content (Ro.TCC): the concentration of total carotenoids in roots determined following the methodology outlined by [27].

### 4.2. Data Analysis

After conducting characterizations across multiple years of cultivation (2011 to 2021, Appendix A), certain qualitative traits exhibited minor fluctuations during various evaluation periods. In response, a data mode approach was implemented to effectively characterize specific cassava accessions, focusing on these nuanced traits. Conversely, for quantitative traits, a rigorous analysis was performed using mixed linear models. This choice was made to address the imbalances inherent in the evaluation tests of cassava germplasm.

To account for the diverse environments and years of cultivation, the combination of the location and year was collectively referred to as “environment” in this study. A joint analysis was conducted, considering all environments for each quantitative descriptor. The analysis was based on the following model: y=Zg+Wb+Ti+e, where *y* is the phenotypic observations; *g* represents the genotypic effects, considered random with distribution as $gN\left(0,\sigma_{g}^{2}\right)$; *b* is the aligned effects of blocks within trials (random effects, $bN\left(0,\sigma_{b}^{2}\right)$; *i* is the effects of genotype × trial interaction (random effects, $iN\left(0,\sigma_{ge}^{2}\right)$; and e represents the error effects (random effects, $eN\left(0,\sigma_{e}^{2}\right)$). The incidence matrices for these effects were denoted as *Z*, *W*, and *T*, respectively. This model was employed to estimate the genetic values of the genotypes by evaluating experiments conducted under an incomplete block design in multiple trials. The analysis of mixed linear models and obtention of the best linear unbiased predictors (BLUPs) was performed using the sommer package in R software version 4.2.2 [47].

### 4.3. Development of the Thematic Collections for Cassava Germplasm

We established three thematic collections, taking into consideration yield traits (CC_Yield), pest and disease resistance (CC_Disease), and root quality traits (CC_Root_quality). The assemblage of the CC_Yield collection involved a process wherein 12 specific clones were chosen based on their low BLUP value for plant architecture and, conversely, their high BLUPs for critical traits such as T.FRY, DRY, Pl.V12M, Pl.V1.5M, DMC, and Ro.NP. Concurrently, the formation of the CC_Disease collection revolved around identifying genotypes with lower BLUPs for mite severity (P.MS), while also exhibiting reduced shoot disease severity like D.AntS, D.CBB, D.BrLS, D.BlLS, D.RS, as well as the root disease D.FS. As for the CC_Root quality collection, the selection process involved genotypes with lower BLUPs for Ro.HCN and Ro.CoT (lower than 25 min). Subsequently, the chosen genotypes underwent an additional refinement stage, emphasizing attributes such as cream/yellow pulp color, with high dry matter and carotenoid content values, along with roots demonstrating a friable or highly friable texture post-cooking.

### 4.4. Kinship Analysis and Overlap Assessment among Cassava Accessions and Thematic Collections

The establishment of thematic collections, as well as the assessment of genetic parameters and kinship analysis, relied on the utilization of phenotypic data and single nucleotide polymorphism (SNP) markers. The SNP marker data, previously reported by [48], was used for grouping the collections and conducting genetic analyses. The SNP data underwent sequence analysis and quality filters using Tassel software version 5.2.37 [49], including the removal of alleles with a minimum frequency (MAF) and the elimination of data with a call rate lower than 5% or greater than 80%. Missing data were imputed using Beagle 4.1 software [50]. A total of 20,023 SNPs distributed across the 18 cassava chromosomes were employed in the subsequent analyses.

Once the collections were established, the extent of kinship between the chosen accessions was gauged using the identity-by-state (IBS) technique on an individual level. This evaluation was executed by employing the snpgdsIBS function from the SNPRelate package [51]. Pairs of accessions displaying a notable degree of kinship (exceeding >0.95) underwent scrutiny, and one individual from each pair was excluded from subsequent analyses. This strategic step was taken to guarantee that the thematic collections featured minimal redundancy among accessions.

To determine the degree of coincidence between the genotypes across the three thematic collections, a binary code approach was employed. Each individual within the collections was assigned a code of 1 if selected or 0 if unselected. Coincidences between the collections were then calculated using the Kappa index [52], which provides a measure of agreement beyond what would be expected by chance alone. These analyses were conducted using R version 4.2.2 [47], utilizing appropriate statistical functions and packages.

### 4.5. The Genetic Parameters of the Thematic Collections

The comparison between different thematic collections and the entire collection for phenotypic data was conducted by examining the dispersion of quantitative and qualitative traits. Shannon–Weaver diversity indices were calculated for each trait in the entire collection and individual collections using the formula $H^{\prime }=-\sum_{i=1}^{n}p_{i}{log}_{e}{(p}_{i})$, where pi represents the observed frequency of class *i* for a given trait, and *n* is the number of the phenotypic classes of the trait. To maintain values between 0 and 1 (ranging from monomorphism to maximum phenotypic diversity), all *H′* indices were normalized and divided by the maximum value. For qualitative traits, *k* denotes the number of classes or scores of the descriptor, while for quantitative traits, six classes were estimated based on the lower and upper limits defined by the total observed range in the entire collection for each trait (Table 4). All analyses were performed using R version 4.2.2 [47].

The genetic diversity within the thematic collections and the entire collection was assessed using a set of 20,023 SNPs. Several parameters were used to evaluate the genetic diversity, including observed heterozygosity (Ho=1−∑k∑iPkii/np), genetic diversity within the population ($Hs=ñ/\left(ñ-1\right)\left[1-\sum_{i}{Pi}^{-2}-\frac{Ho}{2ñ}\right]$), and the inbreeding coefficient (Fis=1−Ho/Hs). All of these parameters were calculated using the hierfstat package [53] of the R version 4.2.2 [47].

### 4.6. The Phenotypic and Molecular Diversity of the Cassava Thematic Collections

The structure of the thematic collections and the entire collection was analyzed using principal component analysis (PCA) on both phenotypic and molecular data. Phenotypic data was analyzed using the AMR package [54], while molecular data was analyzed using the PCAtools package [55], both within the R version 4.2.2 [47].

Furthermore, to discriminate between the cassava thematic collections and the entire collection, a discriminant analysis of the principal components (DAPC) available in the adegenet package for R version 4.2.2 [47], which considered both quantitative and molecular phenotypic data, was employed. The principal components that accounted for more than 80% of the total variance in the data were retained, and cassava genotypes were assigned to their respective thematic collections (CC_Yield, CC_Disease, and CC_Root_quality) based on a priori grouping.

## 5. Conclusions

In this study, three thematic collections were developed focusing on root yield (CC_Yield), resistance to pests and diseases (CC_Disease), and root quality traits (CC_Root_quality). These collections represent approximately 4% of the entire collection. Within each thematic collection, a noticeable improvement in mean values was observed compared to the entire collection. However, it is important to note that there was still a variation of more than 10% in the index of selection within each thematic collection, which was expected, due to the specific nature of each theme. The construction of these collections relied on the ranking of promising genotypes, which proved to be an effective strategy in selecting a diverse set of accessions with the desired traits for each theme.

The three thematic collections presented in this study are characterized by their non-redundancy, manageability, and efficiency in gathering promising accessions, with desired traits related to productivity, resistance to pests and diseases, and root quality. These collections provide a valuable resource for the targeted characterization and organization of selected accessions, thereby facilitating the management and handling of the entire germplasm. Moreover, the carefully selected accessions within these thematic collections serve as a foundation for choosing parental lines that can be utilized in population improvement and in the generation of new cultivars.

## Figures and Tables

**Figure 1 plants-12-03474-f001:**
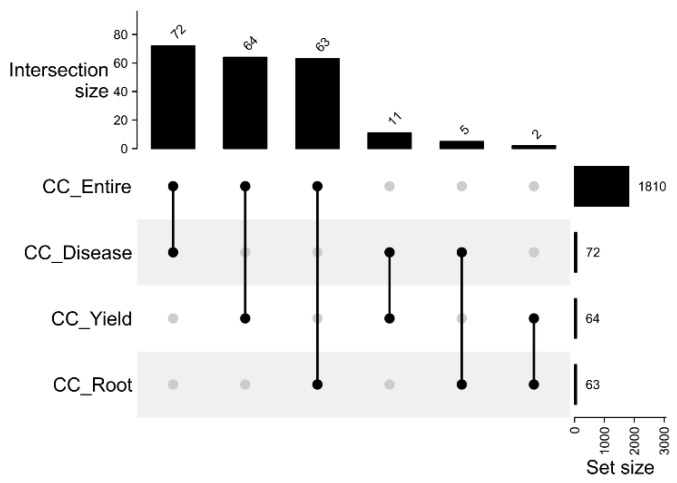
The count of overlapping cassava accessions across distinct thematic collections: CC_Entire—complete collection, CC_Disease—Thematic collection focusing on pest and disease resistance; CC_Root_quality—Thematic collection emphasizing root quality; CC_Yield—Thematic collection centered around root yield traits.

**Figure 2 plants-12-03474-f002:**
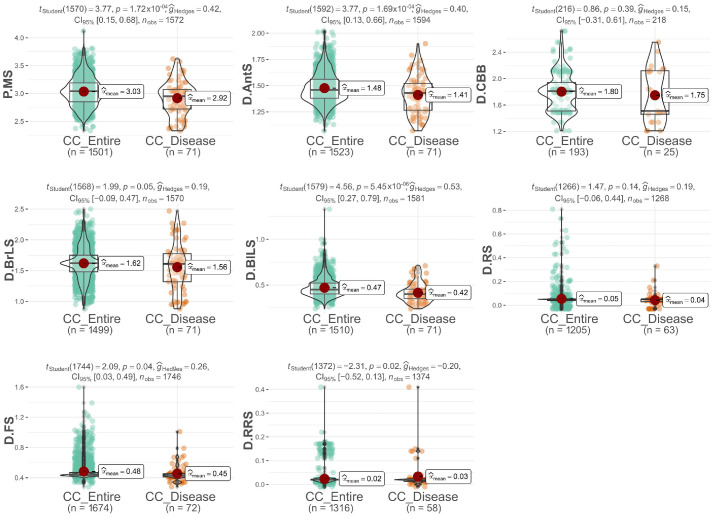
A comparative analysis of various attributes associated with resistance to pests and diseases in cassava. The analysis compares the thematic collection specifically focused on pest and disease resistance (CC_Disease) with the complete collection (CC_Entire). The attributes evaluated include: mite severity (P.MS), anthracnose severity (D.AntS), bacterial blight severity (D.CBB), brown (D.BrLS) and blight leaf spot severity (D.BlLS), rust severity (D.RS), frogskin severity (D.FS), and root rot severity (D.RRS).

**Figure 3 plants-12-03474-f003:**
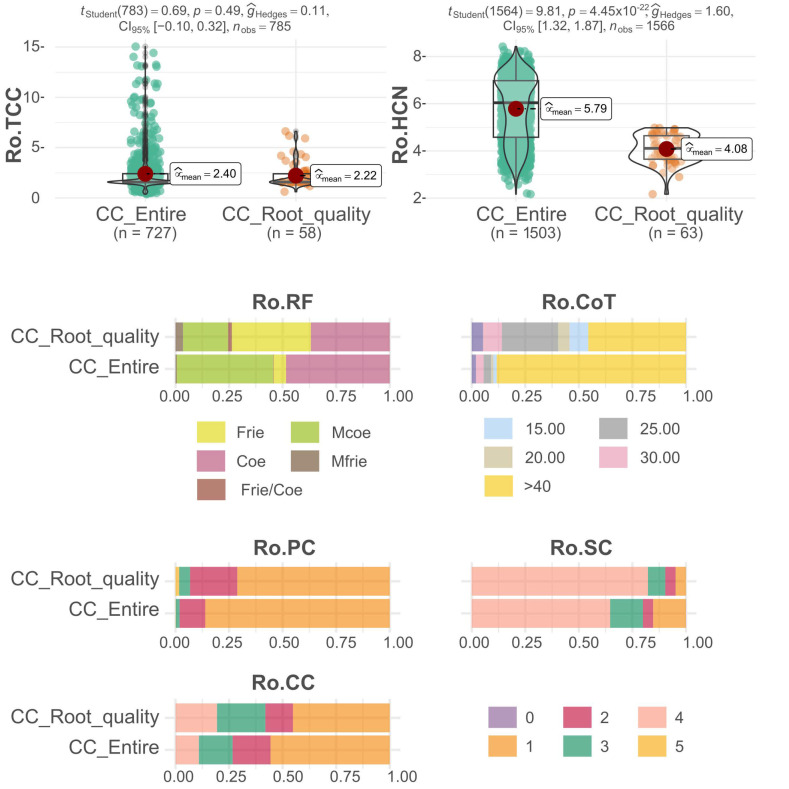
A comparative analysis of different attributes associated with the root quality traits in cassava. It compares the root quality thematic collection (CC_Root_quality) with the complete collection (CC_Entire) of cassava accessions. The analyzed traits include: total carotenoid content (Ro.TCC), hydrogen cyanide content (Ro.HCN), cooked root friability (Ro.RF), root cooking time (Ro.CoT), root pulp color (Ro.PC), root skin color (Ro.SC), and root cortex color (Ro.CC).

**Figure 4 plants-12-03474-f004:**
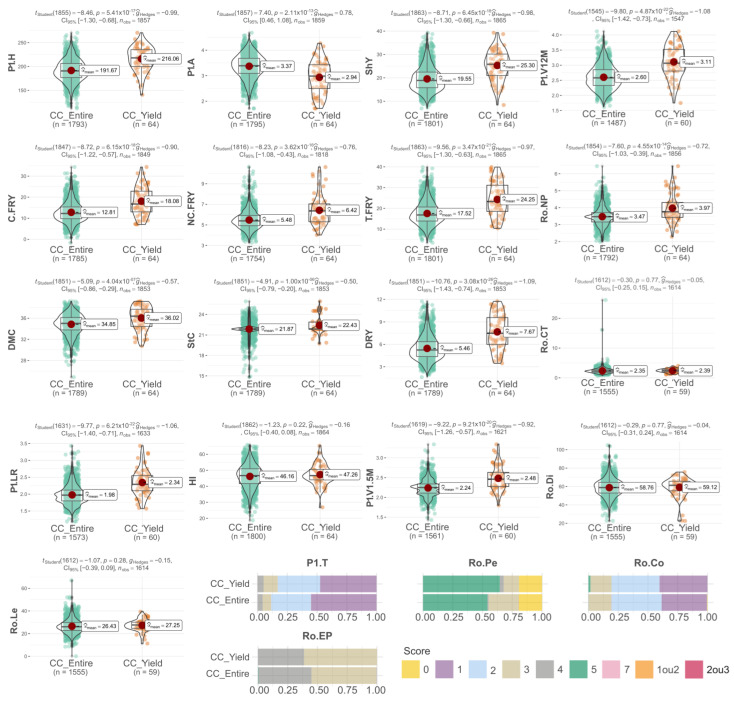
A comparative analysis of yield attributes in cassava genotypes: Thematic collection for yield attributes (CC_Yield) versus the entire collection (CC_Entire) for these traits: plant height (Pl.H), plant architecture (Pl.A), plant type (Pl.T), shoot yield (ShY), marketable fresh root yield (C.FRY), unmarketable fresh root yield (NC.FRY), total fresh root yield (T.FRY), dry matter content (DMC), starch Content (StC), dry root yield (DRY), harvest index (HI), cortex thickness (Ro.CT), presence of peduncle (Ro.Pe), root constriction (Ro.Co), average root length (Ro.Le), average root diameter (Ro.Di), easy root peeling (Ro.EP), leaf retention (Pl.LR), plant vigor at 1.5 months (Pl.V1.5M), plant vigor at 12 months (Pl.V12M), average number of roots per plant (Ro.NP).

**Figure 5 plants-12-03474-f005:**
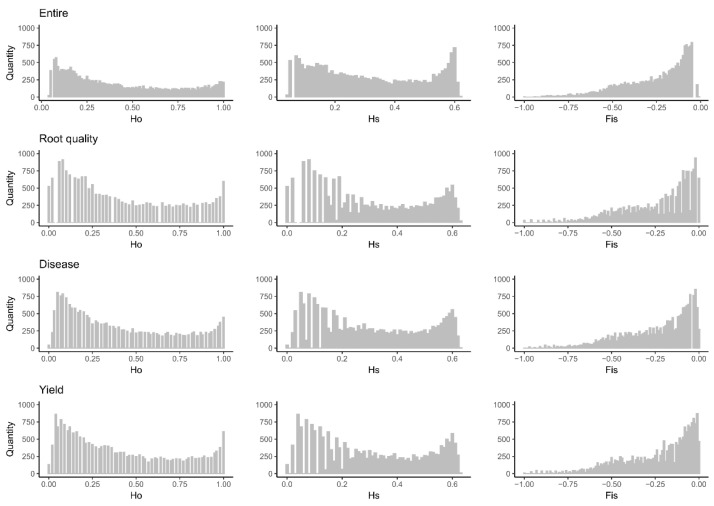
Genetic parameters of the complete collection (CC_Entire) and thematic collections for resistance to pests and diseases (CC_Disease), root quality (CC_Root_quality), and yield traits (CC_Yield), for observed heterozygosity (*Ho*); expected heterozygosity (*Hs*), and inbreeding coefficient (*Fis*).

**Figure 6 plants-12-03474-f006:**
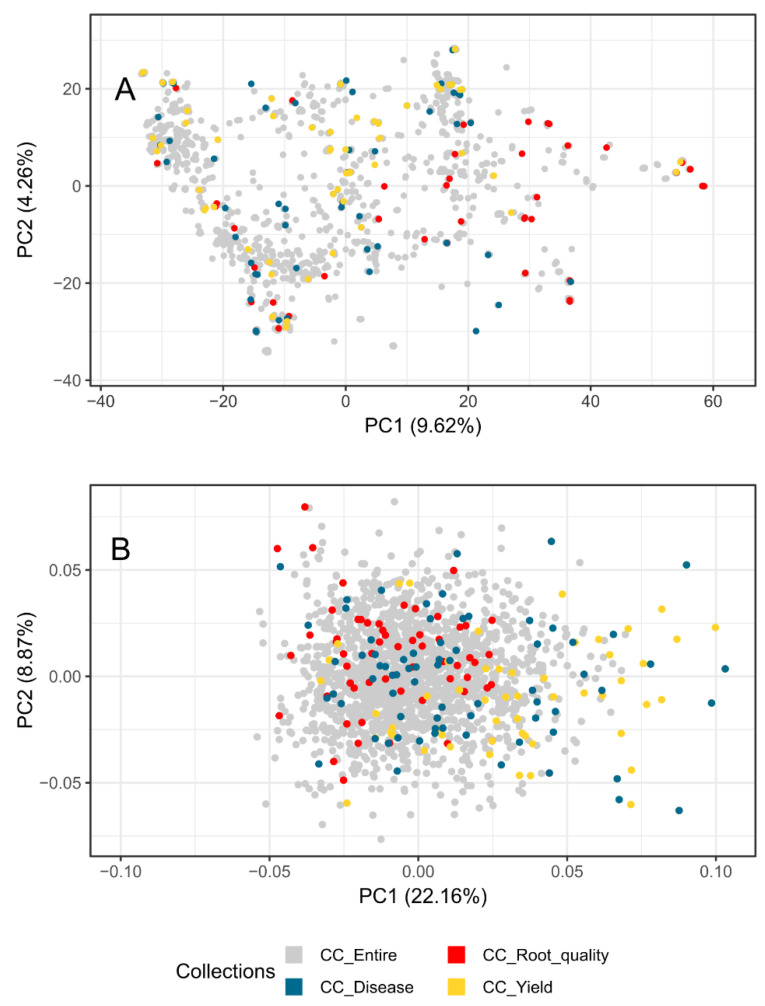
Principal component analysis (PCA) of 1810 cassava accessions based on molecular data from 20,023 SNPs (**A**) and phenotypic data from 27 quantitative traits (**B**). Accessions belonging to specific thematic collections for yield traits (CC_Yield), resistance to pests and diseases (CC_Disease), and root quality traits (CC_Root_quality) are indicated by different colors.

**Figure 7 plants-12-03474-f007:**
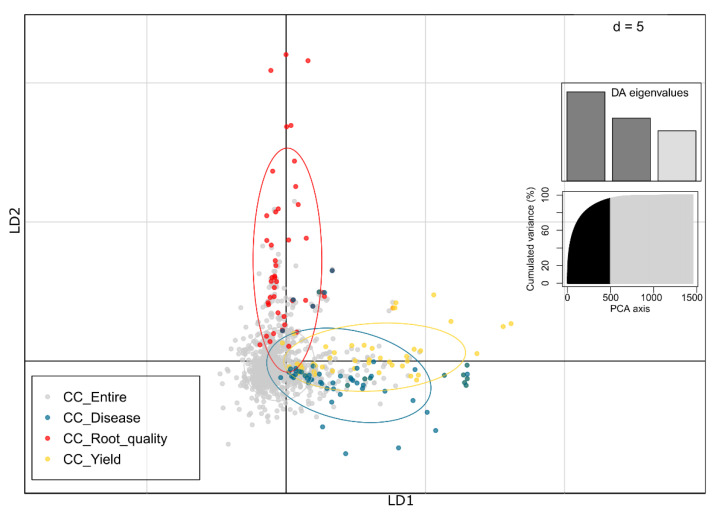
A discriminant analysis of principal components (DAPC) of 20,023 SNPs and 27 quantitative descriptors in 1665 accessions distributed in thematic collections for yield attributes (CC_Yield), resistance to pests and diseases (CC_Disease), and root quality traits (CC_Root_quality).

**Table 1 plants-12-03474-t001:** A Kappa index of coincidence between accessions selected by different cassava thematic collections.

Kappa Index	CC_Disease *	CC_Root_Quality	CC_Yield
CC_Disease	1	0.096	0.176
CC_Root_quality	0.096	1	0.110
CC_Yield	0.176	0.110	1

* CC_Disease—Thematic collection focusing on pest and disease resistance; CC_Root_quality—Thematic collection emphasizing root quality; CC_Yield—Thematic collection centered around root yield traits.

**Table 2 plants-12-03474-t002:** Shannon–Weaver indices of quantitative and qualitative descriptors in three cassava thematic collections, focusing on pest and disease resistance (CC_Disease), root quality (CC_Root_quality), and root yield traits (CC_Yield).

Trait *	Collections				Trait	Collections			
	CC_Entire	CC_Root_Quality	CC_Yield	CC_Disease		CC_Entire	CC_Root_Quality	CC_Yield	CC_Disease
Pl.H	0.73	0.63	0.78	0.77	D.AntS	0.73	0.75	0.65	0.84
Pl.A	0.77	0.75	0.91	0.81	D.CBB	0.76	0.68	0.85	0.85
ShY	0.77	0.61	0.86	0.82	D.BrLS	0.76	0.73	0.62	0.93
C.FRY	0.68	0.56	0.83	0.79	D.BlLS	0.53	0.48	0.50	0.46
NC.FRY	0.61	0.61	0.85	0.67	D.RS	0.12	0.21	0.14	0.15
T.FRY	0.74	0.61	0.89	0.84	D.FS	0.37	0.40	0.30	0.31
DRY	0.77	0.65	0.89	0.87	D.RRS	0.11	0.10	0.24	0.24
StC	0.37	0.71	0.56	0.34	Ro.HCN	0.90	0.47	0.85	0.87
HI	0.75	0.65	0.64	0.73	Ro_TCC	0.38	0.32	0.16	0.34
Ro.CT	0.02	0.05	0.00	0.00	Pl.T	0.64	0.68	0.70	0.60
Ro.Le	0.50	0.49	0.38	0.45	Ro.Pe	0.59	0.58	0.54	0.54
Ro.Di	0.62	0.53	0.61	0.63	Ro.Co	0.56	0.47	0.57	0.55
Pl.LR	0.65	0.54	0.79	0.79	Ro.EP	0.45	0.34	0.42	0.35
Pl.V12M	0.76	0.68	0.92	0.88	Ro.CC	0.84	0.92	0.53	0.74
Pl.V1.5M	0.55	0.53	0.75	0.67	Ro.SC	0.63	0.41	0.51	0.63
DMC	0.66	0.77	0.69	0.64	Ro.RF	0.57	0.78	0.57	0.56
Ro.NP	0.62	0.63	0.87	0.75	Ro.PC	0.34	0.57	0.23	0.29
P.MS	0.73	0.75	0.74	0.78	Ro.CoT	0.30	0.81	0.10	0.18

* plant height (Pl.H), plant architecture (Pl.A), shoot yield (ShY), marketable fresh root yield (C.FRY), unmarketable fresh root yield (NC.FRY), total fresh root yield (T.FRY), dry root yield (DRY), starch content (StC), harvest index (HI), cortex thickness (Ro.CT), average root length (Ro.Le), average root diameter (Ro.Di), leaf retention (Pl.LR), plant vigor at 12 months (Pl.V12M), plant vigor at 1.5 months (Pl.V1.5M), dry matter content (DMC), average number of roots per plant (Ro.NP), mite severity (P.MS), anthracnose severity (D.AntS), bacterial blight severity (D.CBB), brown spot severity (D.BrLS), blight leaf spot severity (D.BlLS), rust severity (D.RS), frogskin severity (D.FS), root rot severity (D.RRS), hydrogen cyanide content (Ro.HCN), total carotenoid content (Ro.TCC), plant type (Pl.T), presence of peduncle (Ro.Pe), root constriction (Ro.Co), easy root peeling (Ro.EP), root cortex color (Ro.CC), root skin color (Ro.SC), cooked root friability (Ro.RF), root pulp color (Ro.PC), root cooking time (Ro.CoT).

**Table 3 plants-12-03474-t003:** The basic parameters of the genetic diversity of thematic collections formed based on different approaches and information from 20,023 SNP markers.

Collections	Observed Heterozygosity (*Ho*)		Genetic Diversity (*Hs*)		Inbreeding Coefficient (*Fis*)		N.º of Alleles
	Mean	Variances	Mean	Variances	Mean	Variances	
CC_Entire	0.403	(0.04–1.00)	0.301	(0.04–0.62)	0.228	(−1.00/ 0.01)	58,671
CC_Disease	0.405	(0.00–1.00)	0.302	(0.00–0.63)	−0.227	(−1.00/ 0.00)	53,251
CC_Root_quality	0.394	(0.00–1.00)	0.292	(0.00–0.63)	−0.226	(−1.00/ 0.00)	51,688
CC_Yield	0.410	(0.00–1.00)	0.303	(0.00–0.63)	−0.234	(−1.00/ 0.00)	52,211

CC_Entire—complete collection, CC_Disease—Thematic collection focusing on pest and disease resistance; CC_Root_quality—Thematic collection emphasizing root quality; CC_Yield—Thematic collection centered around root yield traits.

**Table 4 plants-12-03474-t004:** Estimation of the lower and upper bounds of the phenotypic classes to be used for calculating the Shannon–Weaver diversity index for quantitative traits.

Phenotypic Class	Lower Limit	Upper Limit
1	min	≤min+Amp/6
2	min+Amp/6	≤min+2xAmp/6
3	min+2xAmp/6	≤min+3xAmp/6
4	min+3xAmp/6	≤min+4xAmp/6
5	min+4xAmp/6	≤min+5xAmp/6
6	min+5xAmp/6	≤max

min: minimum value of the trait across the entire collection; max: maximum value of the trait across the entire collection; Amp: range (amplitude) of the trait within the entire collection.

## Data Availability

All datasets generated for this study can be found in the article, Appendix A, and Figshare (https://doi.org/10.6084/m9.figshare.23978346).

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
