# Peer review of "The Development of Thematic Core Collections in Cassava Based on Yield, Disease Resistance, and Root Quality Traits"

_plants, 2023, doi:10.3390/plants12193474_

Round 1

Reviewer 1 Report

The authors presented their work on establishing the thematic core collections using the genebank accession in Embrapa, Brazil. It provided a list of useful materials for germplasm sharing and cassava variety development.

I do not have critical major comments on the manuscript, but I do have several minor comments below.

1. Line 100, please show the full word of IBS here.

2. All the figures and tables: Please double-check the labels in the figures, especially the names of the TC groups and the traits. Please ensure the names in the figures are consistent with the names in the legend and in the manuscript. It is difficult to match the figure with the legend and the manuscript results. Please also pay attention to “_” and “-”

For example, Figure 1, CC_Completa in the figure, and CC_Entire in the legend. Figure 4, I was not able to find the plot I am interested in.

3. Line 98, please briefly describe how the initial selection was performed. Or maybe we can think about the order of the Results sections. How about 2.2 first describing each TC and then 2.1 showing the numbers and overlapping?

4. Results 2.2, Is it possible to mention the quality of the phenotypic data, e.g., repeatability of the measurement or broad-sense heritability? It is important to show the data quality and give the readers confidence about the data and results.

5. Table 3, Please change the headers in English.

6. line 188, please double check the order of the numbers for marketable root yield. Is 18.08 for TC and 12.8 for the entire collection?

NA

Author Response

Reviewer 1

The authors presented their work on establishing the thematic core collections using the genebank accession in Embrapa, Brazil. It provided a list of useful materials for germplasm sharing and cassava variety development.

I do not have critical major comments on the manuscript, but I do have several minor comments below.

  1. Line 100, please show the full word of IBS here.

Response: Ok, thanks.

  1. All the figures and tables: Please double-check the labels in the figures, especially the names of the TC groups and the traits. Please ensure the names in the figures are consistent with the names in the legend and in the manuscript. It is difficult to match the figure with the legend and the manuscript results. Please also pay attention to “_” and “-”

For example, Figure 1, CC_Completa in the figure, and CC_Entire in the legend. Figure 4, I was not able to find the plot I am interested in.

Response: We apologize for the discrepancies that were identified. We have conducted a comprehensive review and have now replaced all the figures with their accurate names.

  1. Line 98, please briefly describe how the initial selection was performed. Or maybe we can think about the order of the Results sections. How about 2.2 first describing each TC and then 2.1 showing the numbers and overlapping?

Response: We added the following sentence:

“In the CC-Yield collection, we selected clones with lower BLUPs for plant architecture and higher BLUPs for other traits. The CC_Disease collection, on the other hand, focused on genotypes with lower BLUPs for mite severity, as well as for diseases affecting both the aerial and root parts of the plant. In the CC_Root_quality collection, genotypes were chosen for exhibiting lower BLUPs in terms of cyanide content and shorter cooking times.”

  1. Results 2.2, Is it possible to mention the quality of the phenotypic data, e.g., repeatability of the measurement or broad-sense heritability? It is important to show the data quality and give the readers confidence about the data and results.

Response: We added Table S2 as supplementary material including this information.

  1. Table 3, Please change the headers in English.

Response: Ok.

  1. line 188, please double check the order of the numbers for marketable root yield. Is 18.08 for TC and 12.8 for the entire collection?

Response: Ok. Thanks.

Reviewer 2 Report

The manuscript with the title "Development of thematic core collections in cassava based on yield, disease resistance and starch quality traits" is an interesting one. The topic addressed is in accordance with the "Plants" Journal.

In order to be published, as a reviewer I would like to make a few observations:

The title mentions "starch quality characteristics", and in the abstract I did not find anything about this aspect. The purpose of this study is not detailed in the abstract.

The same basic information, mentioned above, is also missing from the introduction.

Results and discussions are comprehensively presented. The statistical analysis is carried out meticulously and I appreciate this aspect. Again, I consider that there is often a lack of information about "starch quality characteristics". I don't see any analysis for starch. I recommend adding some kind of result, if you have, if not, I suggest you change the title.

Details about "starch quality characteristics" are also missing from the conclusions, please remedy this non-conformity, as deemed appropriate.

Author Response

Reviewer 2

The manuscript with the title "Development of thematic core collections in cassava based on yield, disease resistance and starch quality traits" is an interesting one. The topic addressed is in accordance with the "Plants" Journal.

In order to be published, as a reviewer I would like to make a few observations:

The title mentions "starch quality characteristics", and in the abstract I did not find anything about this aspect. The purpose of this study is not detailed in the abstract.

Response: We acknowledge that the mention of 'starch quality characteristics' in the title may create incorrect expectations. We would like to clarify that the objective is not directly related to starch, and therefore, we have revised the title to:

“Development of thematic core collections in cassava based on yield, disease resistance and root quality traits"

The same basic information, mentioned above, is also missing from the introduction.

Response: We believe that our modification to the title of the work effectively encompasses the rationale, context, and objectives outlined in the introduction.

Results and discussions are comprehensively presented. The statistical analysis is carried out meticulously and I appreciate this aspect. Again, I consider that there is often a lack of information about "starch quality characteristics". I don't see any analysis for starch. I recommend adding some kind of result, if you have, if not, I suggest you change the title.

Response: We appreciate your feedback and would like to apologize once again for any confusion. We have now modified the title by replacing the word 'starch' with 'root'.

Details about "starch quality characteristics" are also missing from the conclusions, please remedy this non-conformity, as deemed appropriate.

Response: Same previous comment.

Reviewer 3 Report

This manuscript reports on a valuable study to develop thematic collections from within larger germplasm collections for cassava-- among the most important root or underground crops in the tropics. The study followed a well designed and robust methodology and analysis; the manuscript is well written, organized and the data are well presented and discussed.

To provide some perspective to the reader, it would help to provide a brief background on the use of thematic collections for other root crops in the tropics or with cassava, and if not- to highlight that this is among the first studies to do so. 

Additional comments on the text include,

L 22 Re, DAPC, spell out the first time this is mentioned?

L 434-449, Would it help here to include in this paragraph citations on how similar pest thematic collections have been shown to be successful with other crops, or studies?

L 534, Perhaps provide a background on the overall (entire collection). Does it represent global cassava germplasm (inc. Asiaa & Africa), or is it more based on Brazil or Latin America germplasm. Perhaps cite any previous studies that have characterized this collection, or indicate that no previous studies have been conducted on the overall collection.

L 626-627, Perhaps briefly provide some background on the germplasm characterization. Where was this conducted? Any citations on germplasm characterization of this germplasm collection (see comments L 534)?

L 762 & 764, Citations, year of publication?

L 773, Citation, publisher and city of publication?

Minor edit suggestions are included in the attached copy of the manuscript.

////

Excellent English, well written.

Author Response

Reviewer 3

This manuscript reports on a valuable study to develop thematic collections from within larger germplasm collections for cassava-- among the most important root or underground crops in the tropics. The study followed a well designed and robust methodology and analysis; the manuscript is well written, organized and the data are well presented and discussed.

To provide some perspective to the reader, it would help to provide a brief background on the use of thematic collections for other root crops in the tropics or with cassava, and if not- to highlight that this is among the first studies to do so. 

Response: We added the following sentence:

“(...) In this context, extensive efforts have been undertaken to establish core cassava collections, employing diverse methodological approaches. These approaches include the use of molecular markers [16] and, more recently, the integration of quantitative, qualitative, and molecular data, both individually and in combination [17]. These collections encapsulate the broadest phenotypic and molecular diversity among the genotypes within the germplasm bank, while maintaining a limited number of individuals. Nevertheless, thematic collections specifically centered around agronomic attributes have not yet been developed (...).”

Additional comments on the text include,

L 22 Re, DAPC, spell out the first time this is mentioned?

Response: Ok, thanks.

L 434-449, Would it help here to include in this paragraph citations on how similar pest thematic collections have been shown to be successful with other crops, or studies?

Response: We added the following sentence:

“There is a limited literature addressing thematic collections associated with cassava diseases. One of the rare instances in cassava research was carried out by Perez-Fons et al. [14], who dedicated their efforts to establishing a core collection with a focus on enhancing biotic stress tolerance in cassava. In this context, specific traits and metabolites were identified, some of which have the potential to confer intrinsic whitefly tolerance. Conversely, in other species, there are numerous examples of thematic collection formation. In banana, Silva et al. [35] developed a thematic collection comprising accessions intended for the preemptive breeding for Fusarium wilt resistance. Multiple sources of resistance to Fusarium wilt and sterility mosaic disease were identified in the core collection of pigeonpea, both in field and greenhouse conditions [34]. In all of these studies, the establishment of collections facilitated the selection of resistant accessions that will play a significant role in the genetic improvement programs of their respective crops.”

L 534, Perhaps provide a background on the overall (entire collection). Does it represent global cassava germplasm (inc. Asiaa & Africa), or is it more based on Brazil or Latin America germplasm. Perhaps cite any previous studies that have characterized this collection, or indicate that no previous studies have been conducted on the overall collection.

Response: We added this information:

  • “The accessions are from various regions of Brazil, and some obtained through exchange with countries such as Colombia, Venezuela, Nigeria, Mexico, and Uganda. The collection contains local and improved varieties obtained through breeding techniques, including crossings, mass selection, and identification by producers or research institutions.”

  • "In this context, extensive efforts have been undertaken to establish core cassava collections, employing diverse methodological approaches. These approaches include the use of molecular markers [16] and, more recently, the integration of quantitative, qualitative, and molecular data, both individually and in combination [17]. These collections encapsulate the broadest phenotypic and molecular diversity among the genotypes within the germplasm bank, while maintaining a limited number of individuals. Nevertheless, thematic collections specifically centered around agronomic attributes have not yet been developed.”

L 626-627, Perhaps briefly provide some background on the germplasm characterization. Where was this conducted? Any citations on germplasm characterization of this germplasm collection (see comments L 534)?

Response: The information about the geographical location and characterization of the cassava field trials is provided in Table S2, and we've also included additional information in the "Material and Methods" section.

“The accessions are from various regions of Brazil, and some obtained through exchange with countries such as Colombia, Venezuela, Nigeria, Mexico, and Uganda. The collection contains local and improved varieties obtained through breeding techniques, including crossings, mass selection, and identification by producers or research institutions.”

L 762 & 764, Citations, year of publication?

Response: Ok.

L 773, Citation, publisher and city of publication?

Response: Ok.

Minor edit suggestions are included in the attached copy of the manuscript.

Response: Ok.

Round 2

Reviewer 2 Report

The authors responded to all my observations